# Hematological and Extra-Hematological Autoimmune Complications after Checkpoint Inhibitors

**DOI:** 10.3390/ph15050557

**Published:** 2022-04-30

**Authors:** Bruno Fattizzo, Nicolò Rampi, Wilma Barcellini

**Affiliations:** 1Fondazione IRCCS Ca’ Granda Ospedale Maggiore Policlinico, 20122 Milan, Italy; nicolo.rampi@unimi.it (N.R.); wilma.barcellini@policlinico.mi.it (W.B.); 2Department of Oncology and Hemato-Oncology, University of Milan, 20122 Milan, Italy

**Keywords:** checkpoint inhibitors, immune-related adverse events, autoimmune hemolytic anemia

## Abstract

Checkpoint inhibitors (CPI) represent a novel therapeutical strategy with a high efficacy both in solid and hematological cancers. They act by reactivating the immune system against neoplastic cells but may, in turn, cause immune-related adverse events (IRAEs) involving several organs with variable frequency and severity. Up to 10% of CPI-treated patients experience hematological IRAEs, mainly cytopenias. The differential diagnosis is challenging due to underlying disease, previous treatments and the variable liability of available tests (i.e., the direct antiglobulin test, anti-platelet antibodies, etc.). Among extra-hematological IRAEs, cutaneous and endocrine ones are the most frequent (up to 30–50%), ranging from mild (pruritus, eczema and thyroid dysfunctions) to severe forms (bullous disorders, hypophysitis and diabetes), mostly requiring topic or replacement therapy. Gastroenteric and kidney toxicities occur in about 5% of patients, biopsies may support the diagnosis, and immunosuppressive treatment is required in severe cases. Finally, neurologic and cardiologic IRAEs, although rare, may be life-threatening and require prompt intervention. By reviewing the most recent literature on post-CPI IRAEs, it emerged that clinical suspicion and monitoring of laboratory markers of organ damage is pivotal to a prompt diagnosis. In severe cases, CPI should be discontinued and immunosuppressive therapy started, whilst rechallenge is anecdotal and should be carefully evaluated.

## 1. Introduction

Over the years, a better understanding of the relationship between cancer and the immune system has dramatically influenced the treatment of hematological malignancies. In particular, a prominent role has been attributed to the immune escape, by which tumor cells may elude immune surveillance inducing T cell anergy through the activation of surface molecules, namely immune checkpoints.

In this setting, novel drugs, called check-point inhibitors (CPIs), have been developed in order to restore the immune attack against neoplastic cells, achieving impressive clinical outcomes in many cancers. Cytotoxic T-lymphocyte-associated protein 4 (CTLA4), highly expressed on T-regulatory lymphocytes, and the axis of programmed death (PD)1 and its ligand (PDL1) have been widely studied as target of first-generation CPI ipilimumab, nivolumab and pembrolizumab, and more selective drugs were then developed.

These have been licensed for solid tumors (particularly melanoma [1,2], lung [3,4] and renal cancers [5]) as well as for hematological conditions (particularly Hodgkin’s lymphomas [6,7]). In turn, by interacting with immune effectors, excessive stimulation of the immune system may modify the physiological balance between B and T-cells and re-activate effector cells against self-antigens leading to autoimmune manifestations (immune-related adverse events IRAEs) [8]. The latter have been described as the most frequent side effects during CPI treatment and include a wide spectrum of autoimmune manifestations (cytopenias, endocrinopathies, neuropathies, myositis, colitis and nephritis).

Considering all IRAEs, a higher incidence has been documented more frequently in patients treated with anti-CTLA4 antibodies, such as ipilimumab as compared with anti-PD1/PDL1 as pembrolizumab and nivolumab, with an increased risk registered in the case of combination therapy with two CPIs. Since cancer patients are considered frail by definition, the development of such complications may further burden the disease outcome.

Additionally, the differential diagnosis of immune-related complications will be challenged by the underlying neoplastic disease and by toxicities derived from previous treatments. In particular, cytopenias are highly frequent among cancer patients, and thus the recognition of hematological IRAEs, such as autoimmune hemolytic anemia (AIHA), may be delayed. Finally, IRAEs may be life-threatening and should be therefore rapidly diagnosed to allow prompt intervention and subsequent monitoring [9]. In this article, we will review the available literature regarding IRAEs developing after CPIs with a particular focus on hematological ones.

## 2. Hematological Autoimmune Complications

Hematological toxicities have been more frequently reported during anti-PD1 administration and mostly described in the form of unilinear or bilinear cytopenias (Table 1) [8,9]. The latter may have several causes in cancer patients, including the inhibitory effect of cancer-associated inflammation on hematopoiesis, bone marrow metastasis, the toxic effect of chemo- and radiotherapy and the induction of an autoimmune attack against hematopoietic precursors and peripheral blood cells.

All of these represent differential diagnosis of IRAES that are thought to result from a tolerance break induced by the T-cell activation after CPIs. From a meta-analysis, including 9324 patients, incidences of 9.8%, 2.8% and 0.94% emerged for anemia, thrombocytopenia and neutropenia, respectively [9], with AIHA as the most frequent complication, occasionally characterized by a fulminant course.

### 2.1. Autoimmune Hemolytic Anemia

AIHA is caused by autoantibodies against red cells and is classified by the direct antiglobulin test (DAT or Coombs test) into warm, cold and mixed forms in accordance to the thermal range of the autoantibody and to its isotype [10]. The warm variant (approximately 48–70% of all cases) is generally associated with IgG autoantibodies with thermal range of about 37 °C, whereas cold AIHA (nearly 15–25% of cases) is usually caused by IgM autoantibodies with thermal range between 4 and 34 °C.

Furthermore, mixed AIHA and AIHA with a negative DAT have also been described [10,11]. The distinction of the warm forms from the cold ones is fundamental because different treatments are required. In the case of warm AIHA, steroids represent the first line therapy, followed by the anti-CD20 monoclonal antibody rituximab with response in 70–80% of the patients. When dealing with cold AIHA, steroids are effective only at high unacceptable doses, and rituximab should be administered as first line with 50–60% of responses, mainly partial response and short-lasting.

A recent revision of FDA registers documented 68 cases of AIHA developing after CPI [12] with no gender differences and mainly arising in patients with melanoma (41%) and small cell lung cancer (26%), followed by renal cancer, Hodgkin’s lymphoma and non-melanoma skin cancer. Most cases occurred in North America (49%) and Europe (34%), whereas only a minority was in Asia and Australia (10% and 7% respectively). Among different CPIs, 43 cases developed after nivolumab, 13 after pembrolizumab, 7 after ipilimumab and 5 after atezolizumab, with a total of 11 episodes occurring after combination treatment with two CPIs.

The median time of AIHA onset was 50 days from CPI start, and 11 patients had accompanying inflammatory manifestations (four with thrombocytopenia, four with endocrinopathies and three with gastroenteric toxicities). Most patients presented with IgG-positive warm AIHA, whilst cold forms were more rarely described. Almost all episodes were severe (Hb < 8 g/dL) with 80% of patients requiring transfusion support. Similarly, in another recent analysis of 14 AIHA cases after CPI exposure, median time from CPI to AIHA onset was 55 days (interquartile interval IQR 22–110 days) [13].

In comparison to primary AIHA, these patients presented a higher proportion of negative Coombs test (38%) and more severe anemia (median Hb of 6.3 g/dL, IQR 6.1–8 g/dL). Moreover, 50% of patients experienced a relapse after first line therapy, and 14% became chronic. From a therapeutic point of view, CPI discontinuation was necessary in all cases, and the administration of steroids (usually prednisone 1.5–2 mg/kg day or equivalent) was started.

In relapsed/refractory patients, early use of rituximab was adopted [3,4,6,7] and, in the case of hyper-hemolytic manifestations, transfusion support, intravenous immunoglobulins (IVIG) and plasma exchange (PEX) were also useful. One of the main discussed issues is the rechallenge with CPI after complete resolution of AIHA. In this regard, there is a case report of a patient affected by Hodgkin’s lymphoma developing AIHA after nivolumab therapy. The patient responded to steroids and had no relapse after rechallenging with CPI [14].

The mortality of CPI-related AIHA may reach 17% and it is mainly related to multiorgan failure as a fatal consequence of misdiagnosis and late recognition. In fact, the differential diagnosis of anemia in such heavily pretreated and frail patients remains challenging, especially because of the high proportion of Coombs negative cases. As summarized in Figure 1, the suspicion of AIHA should be raised in patients receiving CPIs who display Hb decrease along with altered hemolytic markers (which should be therefore monitored in these patients).

Polyspecific and monospecific DAT should be promptly performed, and more sensitive tests should be asked to the transfusion center in the case of negativity. CPI should be discontinued and steroid treatment started, both in DAT positive cases and in negative ones, once excluded other causes of hemolytic anemia. Supportive measures with transfusions, IVIG, and PEX should be taken into account according to the severity of the clinical picture, and the administration of recombinant erythropoietin (e.g., epoetin alpha 40,000 UI weekly subcutaneously) is a valid option to support hemoglobin response in patients with inadequate reticulocytosis [15]. Finally, early rituximab should be considered in patients not responding to high dose steroids during the first 7–15 days of treatment.

### 2.2. Other Hematological Toxicities

Hematological immune-related IRAEs other than AIHA have been reported in less than 0.6–1% of patients treated with CPI [16,17]. Although rare, these manifestations are associated with a relatively high mortality up to 14% due to possible complications. Immune thrombocytopenia (ITP) and idiopathic neutropenia [18] are the most frequent, variably occurring from 10 [19] up to 25 weeks [16] from CPI.

**Table 1 pharmaceuticals-15-00557-t001:** Hematological toxicities after checkpoint inhibitors (CPI).

References	Type of Study	Patients	Frequency	Main Findings	CPI Interruption
**Delanoy N et al. (2019)** [17]	Observational study	745	3.7%	The most-frequent hematologic IRAEs after anti-PD-1 or anti-PD-L1 were AIHA, ITP or neutropenia (26%), followed by pancytopenia or aplastic anemia (14%). The median time of onset was 10 weeks; most events were grade 4 and resolved after immunosuppressive therapy.	80% of the cases 20% rechallenge
**Michot JM et al. (2019)** [19]	Review	63	3.6%	An incidence of 0.7% for grades 3 to 4 IRAEs, mostly immune cytopenias (17 to 29%), aplastic anemia (19%) and HLH (11%). The median time of onset was of 10 weeks. Resolution varied from 25% for aplastic anemia to 80% for ITP and AIHA, and 14% died. The risk of recurrence after CPI rechallenge was around 50%.	Not reported
**Davis E.J. et al. (2019)** [20]	Observational study	164	1% (among all reported adverse events)	AIHA was the most common, mostly associated with melanoma and lung cancer; 23% had an extra-hematological IRAEs; mortality was 11% but increased to 23% in the case of HLH.	Not reported
**Zaremba A. et al. (2021)** [18]	Observational study	6961	0.14%	10 patients experienced grade 4 neutropenia (60% possibly due to metamizole), with median time of onset of 6.4 weeks; 40% required systemic steroids, and neutropenia responded to G-CSF. No recurrence was reported after CPI rechallenge.	70%
**Kramer R et al. (2021)** [16]	Observational study	7626	0.6%	Mostly autoimmune cytopenias (28–34%), rarely HLH (4%), aplastic anemia (2%), coagulation dysfunction (2%) and acquired hemophilia A (2%). The median time of onset was 25 weeks. 60% required hospitalization, and 80% had complete resolution. AIHA and ITP tended to persist.	60%

IRAEs immune-related adverse events, ITP: immune thrombocytopenia, AIHA: autoimmune hemolytic anemia, HLH: hemophagocytic lymphohistiocytosis, G-CSF: granulocyte colony stimulating factor.

Aplastic anemia, including some cases of pure red cell aplasia, and hemophagocytic lympohohistiocytosis (HLH) are even rarer and are generally characterized by poor outcome with a mortality rate of 23% for HLH in a recent report [20]. Given their autoimmune/autoinflammatory nature, these manifestations are thought to be due to CPI-mediated immune dysregulation. A presumptive hyperinflammatory state as a result of the inhibition of cytotoxic T-cell activity appears to be the background of CPI-related HLH [21].

Additionally, concomitant medications, such as metamizole [18], may induce cytopenias through idiosyncratic reactions as a consequence of immune imbalance. Overall, such forms may be difficult to diagnose and often require the exclusion of other secondary forms, possibly delaying proper treatment. Differently from AIHA, where DAT is available for the diagnosis, other hematologic IRAES are diagnosed after observing a drop of blood counts or alteration of HLH markers (serum ferritin, triglycerides, cytopenias, organomegalies, fever, etc.).

Cumulatively, hematological IRAEs required hospitalization in more than half of cases, due to grade 3–4 events in about 76% of patients [17]. Nevertheless, most cases (about 80%) showed complete recovery, whilst ITP, similarly to AHIA, showed a higher frequency of persistency [16]. Given the severity of these IRAEs, holding CPI therapy is advised starting from grade 2 toxicities, with permanent discontinuation in the case of grade 4 ones [19]. Furthermore, rechallenge with CPI requires special attention, since the risk of recurrence has been estimated to be as high as 50%. Altogether, even if rare, these complications are potentially fatal and require high awareness and adequate clinical and laboratory monitoring to establish prompt therapeutic measures [17].

## 3. Extra-Hematological Toxicities after Checkpoint-Inhibitors

Beyond hematologic ones, autoimmune manifestations during CPI therapy may involve several systems and organs. Hereafter, we will describe the main complications according to their incidence (Figure 2).

### 3.1. Immune-Related Endocrinopathies

Among extra hematological toxicities, endocrinopathies (Table 2) embody one of the most common IRAEs associated with CPIs, representing 30–35% of cases as reported in a recent meta-analysis from Rubino et al. [22]. Clinically, most cases may be classified in three different nosocomial entities: hypophysitis, mostly associated with anti-CTLA4 antibodies, thyroid dysfunction, more related to anti-PD1 CPIs and insulin-deficient diabetes mellitus. About the former, in a meta-analysis, Barroso-Sousa and colleagues [23] reported a high incidence in patients receiving ipilimumab. After a median of 150 days from the first infusion, patients generally complained cephalalgia, and brain MRIs documented pituitary enlargement [24].

Contrarily, hypophysitis occurring after anti-PD1 drugs seems to have a later presentation with more heterogeneous symptoms (fatigue, loss of appetite and myalgias/arthralgias) and no MRI alterations [25,26]. Nevertheless, testing for hormone deficiencies and subsequent supplementation are warranted, not only of the adrenocortical axis but also thyroid hormones [26]. Regarding the latter, thyroid dysfunction may occur in up to 30% of patients treated with anti-PD1 inhibitors [22]. After an initial transient thyrotoxic phase, with an earlier onset than other drug-related thyroiditis, most subjects experienced hypothyroidism, mostly with no need for hormone replacement [27].

Pre-existing thyroid dysfunction represents a significant risk factor for the development of such toxicity [28], whilst the role of anti-TPO autoantibodies is controversial (no association with the time to thyroiditis but with more severe thyroid dysfunction) [29,30]. Regarding CPI-related diabetes-mellitus, a total of 144 cases have been reported, and Quandt and colleagues [31] described islet-antibodies positivity in only 49% of them. Most patients had been treated with anti-PD1 inhibitors [31], and the median time to complication onset was of 7–17 weeks (shorter in severe cases).

Importantly, all patients reported in a Canadian cohort remained insulin-dependent at the end of follow-up [32], thus, highlighting a chronic damage despite treatment with steroids and infliximab in refractory patients. Overall, endocrine IRAEs represent a common finding in patients treated with CPIs, warranting monitoring. Although mainly mild, they may be potentially irreversible and often require replacement therapy. Early discontinuation of CPI should be considered in the case of grade 3 or 4 toxicities.

**Table 2 pharmaceuticals-15-00557-t002:** Immune-related endocrinopathies after checkpoint inhibitors (CPI).

References	Type of Study	Patients	Frequency	Main Findings	CPI Interruption
**Faje AT et al. (2014)** [24]	Observational study	154	11%	Immune hypophysitis in melanoma patients after Ipilimumab in a dose-dependent manner. Brain MRI may detect pituitary enlargement in symptomatic patients. Hormone deficiencies may persist.	Not reported
**Morganstein D. et al. (2017)** [27]	Observational study	191	23% with anti-CTLA4, 39% with anti-PD-1 50% if in combination	Thyroid IRAEs occurred after a median of 30–60 days, more frequently in males. A hyperthyroidic phase followed by hypothyroidism is mainly observed. Altered TSH before treatment may be a predictor.	Not reported
**Osorio J. et al. (2017)** [29]	Observational study	51	21%	Lung cancer patients treated with pembrolizumab with anti-thyroid antibodies were at higher risk of thyroid IRAEs. A biphasic pattern (hyperthyroidism followed by hypothyroidism) was described and replacement therapy was needed.	0%
**Garon-Czmil J. et al. (2019)** [26]	Observational study	249	37% among endocrine ir-AEs	Hypophysitis was more frequent with Ipilimumab, after 80 to 160 days; brain MRI may show pituitary enlargement. Nearly all patients required hydrocortisone supplementation (90%) and 20% thyroid hormones.	1 patient
**Faje, A. et al. (2019)** [25]	Observational study	22	0.5% anti-PD1 13.6% anti-CTLA4	Hypophysitis developed after 77 to 500 days. Symptoms were more subtle after anti-PD1 (fatigue, loss of appetite and myalgias/arthralgias) versus anti-CTLA4. Brain MRI was not informative.	5 patients
**Presotto E.M. et al. (2020)** [28]	Observational study	179	30.2%	Thyroid alterations occurred in 29.6%. Pre-existing thyroid dysfunction was a risk factor. IRAE occurred within 2 months and 75.5% of cases required replacement therapy.	Not reported
**Kotwal A. et al. (2020)** [30]	Observational study	91	25%	TPO antibodies were detected only in 22% of patients with thyroid IRAEs. Higher TPO titer may be related to more severe thyroid dysfunction. Longer time from thyrotoxicosis to hypothyroidism was described as compared to other thyroid disorders.	0%
**Quandt, Z. et al. (2020)** [31]	Review	53	0-2-1.4%	Diabetes mellitus was most frequent with anti-PD1/PD-L1, after 7–17 weeks (shorter in patients with anti-islet antibodies). Steroids worsened insulin resistance.	Not reported
**Rubino R. et al. (2021)** [22]	Observational study	251	27.89%	Thyroid IRAEs were the most frequent and may be predicted by pre-existing endocrinopathy. Female were more affected and required replacement in 45%. A correlation between IRAEs and a better outcome (PFS and OS) was reported.	25%
**Muniz et al. (2021)** [32]	Observational study	34	Not reported	Diabetes mellitus developed after a median of 2.4 months and was more frequent with anti-PD1/PDL1. 62% of patients had an acute onset with ketoacidosis with a mortality of 5%, and some became chronic. All patients were treated with insulin therapy and in 12% with immunosuppressive therapy.	56%

TSH: thyroid-stimulating hormone, MRI: magnetic resonance imaging, NSCLC: non-small cell lung cancer, TPO: thyroid peroxidase, PFS: progression-free survival, OS: overall survival, DKA: diabetic ketoacidosis, DM: diabetes mellitus.

### 3.2. Cutaneous Adverse Events

Cutaneous IRAEs are another frequent adverse event after CPIs (Table 3), affecting 30% to 50% of treated patients [33]. Clinically, cutaneous IRAEs could be divided into three broad categories according to the common terminology criteria for adverse events (CTCAE) classification [34]: (1) inflammatory eruptions as described by Coleman et al. [35], including lichenoid, eczematous psoriasiform reactions and maculopapular drug exanthems; (2) bullous dermatoses, usually described with a latency longer than that of other cutaneous toxicities [36]; and (3) cutaneous sarcoidosis and vitiligo-like depigmentation rash.

Overall, pruritus accounts for the most frequent symptom that may precede or represent itself as a dermatological AE [37,38]. Regarding etiopathogenesis, further proof of the autoimmune attack comes from melanoma patients where an overexpression of melanoma-associated antigens on cutaneous cells have been demonstrated. The latter may be targeted by the immune system enhanced by the CPI [39]. Interestingly, this off-target hyper-immune effect correlates with a longer progression free survival (PFS) in patients experiencing cutaneous IRAEs [40].

Furthermore, Matsuya et al. [39] reported how the progression of vitiligo to involve a broader body surface represents a sensitive predictive factor of durable tumor response and prolonged PFS after anti-PD-1. Cutaneous manifestations frequently appear about 5 months after the start of the drug and may be efficiently controlled with either topic or systemic therapy, mainly steroid-based in the case of high grade IRAEs. Contrarily to hematological IRAEs, most authors advise not to stop treatment with CPI due to cutaneous AE, especially in the setting of advanced disease, since, although common, they are usually manageable.

**Table 3 pharmaceuticals-15-00557-t003:** Cutaneous immune-related adverse events after checkpoint inhibitors (CPI).

References	Type of Study	Patients	Frequency	Main Findings	CPI Interruption
**Naidoo J et al. (2016)** [38]	Case series	3	Not reported	Bullous Pemphigoides (BP) on anti–PD1/PDL1 inhibitors may occur after several months and may be accompanied or preceded by pruritus. Discontinuation of CPI may not determine resolution of BP.	100%
**Hwang SJ et al. (2016)** [33]	Observational study	82	49%	Cutaneous IRAEs included lichenoid reaction (17%), eczema (17%) and vitiligo (15%) in melanoma patients with anti-PD1/PD-L1.	Not reported
**Siegel J et al. (2018)** [36]	Observational study	853	1%	BP occurred after anti-PD1/PDL1 CPIs and had mucosal involvement in 30%; may be determined by autoantibodies against hemidesmosome protein BP180. Steroids were recommended if > 30% of body surface was involved.	1%
**Lee YJ et al. (2019)** [37]	Observational study	211	16,4%	Pruritus was reported as the main manifestation, followed by eczema and maculopapular rash, after a median onset of 50 days. Longer PFS may occur in such patients.	0%
**Chan L. et al. (2020)** [40]	Observational study	82	40%	Cutaneous IRAEs occurred after a median of 6 months. Longer PFS may occur in patients experiencing cutaneous IRAEs	Not reported

BP: bullous pemphigoides.

### 3.3. Gastroenteric Side Effects

Even gastroenterological IRAEs (Table 4) may be commonly observed during CPIs and mainly include diarrhea and colitis. Higher rates have been reported in patients treated with anti-CTLA4 antibodies compared to anti-PD1 for both diarrhea and colitis (30.2–35.4% vs. 12.1–13.7% and 5.7–9.1% vs. 0.7–1.6%, respectively) [41]. This may be explained by the demonstration that anti-CTLA-4 antibodies may abolish T cell-mediated protection to commensal bacteria inducing overactivation of T-cell effectors [41]. Nearly half of patients presented with grade 3 diarrhea, occurring after about 2 months from starting CPI [42].

By endoscopy, an ulcerative pattern was detected in nearly one third of patients [43,44] mostly localized in the descending colon. Data on potential biomarkers that may identify patients at high risk of such IRAEs are controversial: calprotectin levels and qualitative lactoferrin correlated with endoscopic and histological findings in a study by Abu-Sbeih et al. [43]; however, this was not confirmed in a report by Cheung et al. [45]. In any case, an early endoscopic evaluation appears useful to identify high risk features [42], since a correlation between endoscopic scores and clinical outcomes has been reported, and a more intensive immunosuppression is advisable in the case of active inflammation [45].

In this setting, the discontinuation of CPI and administration of steroid therapy have been quite effective, obtaining a clinical remission in almost all cases despite a high recurrence rate. In non-responders, good outcomes have been reached with anti-TNFα drugs, such as infliximab and vedolizumab, usually reserved for very severe cases. Rechallenge with CPI may reactivate colitis. However, Geukes et al. [42] described that the pre-emptive use of vedolizumab at rechallenge was characterized by a low number of recurrences versus CPI alone.

Finally, CPI-related hepatitis should be mentioned. An acute liver failure occurring after a median of 12 weeks was described in 7.7% of the cases. Biopsy is recommended to exclude other possible causes, especially after nonresponse to first line therapy. Unlike autoimmune hepatitis, no specific biomarkers nor characteristic histologic findings have been identified [46], and a third of patients required the addition of a second immunosuppressant drug. Overall, along with liver function tests, prompt endoscopic investigation is recommended in patients treated with CPI developing gastrointestinal symptoms, since histologic findings are fundamental for the differential diagnosis and may suggest stronger immunosuppressive treatment in severe cases.

**Table 4 pharmaceuticals-15-00557-t004:** Gastroenteric toxicities after checkpoint inhibitors (CPI).

References	Type of Study	Patients	Frequency	Main Findings	CPI Interruption
**Abu-Sbeih H et al. (2018)** [43]	Retrospective study	182	43% grade 3/4 diarrhea 32.4% grade 3/4 colitis	Grade 3 colitis affected mostly left colon; at endoscopy one third showed ulcerative pattern. 77.5% patients required immunosuppressant treatment. All patients reached clinical remission and 30% histological remission. The recurrence of colitis occurred in 28% of subjects.	66%
**Geukes Foppen et al. (2018)** [42]	Systematic review and meta-analysis	92	56% with anti-CTLA4 22% with anti-PD1	In 44% of cases, diarrhea was grade 3 and 30% had ulcers at endoscopy; half of patients was refractory to steroids and required Infliximab. The presence of ulcers and pancolitis (≥3 affected colon segments) predicted refractoriness to steroids.	Not reported
**Cheung et al. (2020)** [43]	Retrospective study	134	10%	Higher risk of colitis with combination therapy (anti-PD1/PD-L1 and anti-CTLA4 inhibitors). No predictors; 23% of patients were rescued with Infliximab due to erosions; earlier administration does not seem beneficial.	Not reported
**Bellaguarda et al. (2020)** [41]	Systematic review	Not reported	30.2–35.4% after anti-CTLA4 12.1–13.7% after anti-PD1	The median onset time of gastroenteric toxicities was 4 weeks with anti-CTLA4 and 2–4 months with anti-PD1/PD-L1. Supportive therapies, CPI discontinuation, systemic steroids (effective in 85% of patients) and biological drugs (Infliximab and vedolizumab) were used.	Grade 3 temporarily discontinuation Grade 4 permanently discontinuation
**Riveiro-Barciela et al. (2020)** [46]	Retrospective study	414	6.8%	Severe hepatitis resulted in acute liver failure in 7.7% of cases. Mostly related to anti-PD1/PD-L1 agents, after a median of 12 weeks. All were treated with steroids, and 35.7% required a second line. No recurrence after CPI rechallenge.	100%

### 3.4. Neuromuscular Complications

Neuromuscular IRAEs (Table 5) are rare, with an estimated incidence of 2.9–4.2% in anti-PD1 treated patients, although potentially life-threatening. The onset of neurological IRAEs generally occurs variably between the third [47] and the sixth CPI infusion [48]. Muscular involvement is often described in terms of myositis and myopathies, particularly as myastenia gravis (MG) but also as peripheral polyneuropathies, such as Guillain–Barrè (GB)-like syndrome. The former presents clinically with oculomotor and bulbar signs at higher rate as compared to idiopathic MG [47] and, when tested, autoantibodies traditionally associated with MG may be detected in 40% to 50% of patients [47,49].

Furthermore, in subjects treated with anti-PD1, the association of MG and myopathies has been also described [47,49]. This may result in respiratory complications associated with increased mortality, and the serial monitoring of rhabdomyolytic indexes, such as creatin phosphokinase, which are incremented in the case of myopathies and not in MG, are very useful. Another potentially life-threatening complication is myocarditis, which may be associated with myositis in nearly one third of cases [48] and should be promptly diagnosed.

Concerning other biomarkers, Möhn et al. [50] documented increased blood cell counts in the cerebral-spinal fluid (CSF) of 60% of CPI-treated patients with neurological symptoms, even in the contest of Guillain–Barrè like syndrome. Finally, only 28 cases of post-CPI encephalitis with negative CSF examination have been reported in the literature, and they were characterized by a mortality rate of 18% [51]. Since these complications are rare but potentially life-threatening, clinicians should maintain high awareness, and prompt investigations should be conducted at the minimal suspicion. Importantly, an early discontinuation of CPI and immunosuppressive medication initiation should be considered, because neurological side effects may often be irreversible [52].

**Table 5 pharmaceuticals-15-00557-t005:** Neuromuscular toxicities after checkpoint inhibitors (CPI).

References	Type of Study	Patients	Frequency	Main Findings	CPI Interruption
**Möhn, N et al. (2019)** [50]	Systematic review	81	3.8% with anti-CTLA4 6.1% with anti-PD1	Myasthenia and Guillain–Barrè syndromes (GBS) were the most common, followed by peripheral polyneuropathies. Complete response occurred in 37.2% of cases.	Not reported
**Galmiche S et al. (2019)** [51]	Case series	5	Not reported	Encephalitis manifested with headaches, confusion, ataxia, anisocoria and/or dysarthria and meningeal symptoms, with negative CSF and brain MRI findings. The median time of onset was 42 days and required early discontinuation of CPI and prompt immunosuppression. Mortality rate reached 18%.	100%
**Liewluck T. et al. (2018)** [49]	Observational study	654	0.76%	Pembrolizumab-related myopathies mostly affected oculobulbar muscles. AChR antibodies were detected in 50%. Overall, non-necrotizing myopathy responded well to immunosuppressive therapies. Evaluation of myocardium involvement is recommended.	100%
**Moreira A. et al. (2019)** [48]	Observational study	38	Not reported	Myositis occurred at median of 19 weeks after CPI start, often with oculomotor symptoms and usually preceded by other IRAEs. Myocarditis was present in 32% of cases with increased CPK in 43% of patients. 50% responded to steroids and 2 patients died.	50% permanently stopped 25% interrupted
**Johansen A. et al. (2019)** [47]	Systematic review	85	Not reported	Myastenia Gravis (27%), neuropathy (23%, mostly Guillain–Barrè syndrome) and myopathy (34%) were the most frequent. The median time of onset was of 3.6 cycles of anti-PD1/PD-L1 inhibitors. Ach-R antibodies were detected in 50% of patients. 79% responded to steroids.	Not reported

GBS: Guillain–Barrè syndrome, MG: Myasthenia Gravis, CSF: cerebrospinal fluid, AChR: acetylcholine receptor, CPK: creatinphosphokinase.

### 3.5. Nephrotoxicity

Though infrequent, the incidence of grade 3–4 acute kidney injury (AKI) post-CPI ranged from 2 to 5% in clinical trials [53] and mainly developed after 15 weeks from the start of CPI [53,54] (Table 6). Pathologically, an acute tubulointerstitial nephritis was proved to be the main lesion. In a systematic review by Kitchlu et al. [55] including 45 patients with biopsy-confirmed AKI, nearly one third of cases were attributed to pauci-immune glomerulonephritis and renal vasculitis [56] (27%), followed by podocytopathies (including minimal change diseases and focal segmental glomerulosclerosis) and, lastly, C3 glomerulonephritis. Clinically, acute nephrotoxicity may manifest either as nephrotic or nephritic syndrome, with sub-nephrotic proteinuria as the most common urine finding [57].

A proportion of patients, ranging from 7% [53] to 25% [54], needed renal replacement therapy, and a subgroup became dependent from dialysis. Regarding acute tubulointerstitial nephritis, previous and concurrent treatments, including proton pump inhibitors (PPI), may represent potential risk factors in onco-hematological patients. In this view, Cortazar and colleagues [53] identified three different risk factors for AKI development: concomitant use of PPI [58], lower glomerular filtration rate at baseline and concomitant use of anti-PD1 and anti-CTLA4 drugs.

Therapeutically, the prompt discontinuation of CPIs, the evaluation and correction of possible confounders and the early establishment of systemic steroid treatment may lead to a complete restoration of kidney function in most cases. Recurrence occurred in 23% of rechallenged patients [53,54]. Overall, the monitoring of kidney function is also advisable during CPI treatment, and early renal biopsy is useful in those developing AKI to differentiate CPI-related AKI from acute tubular injury due to previous chemotherapy, thereby, avoiding trivial CPI discontinuation.

**Table 6 pharmaceuticals-15-00557-t006:** Nephrotoxicities after checkpoint inhibitors (CPI).

References	Type of Study	Patients	Frequency	Main Findings	Stop CPI
**Shirali A. et al. (2016)** [58]	Case series	6	Not reported	Consider concomitant therapies that may cause idiosyncratic AKI (PPI and NSAID).	100%
**Gallan A.J. et al. (2019)** [56]	Case series	4	Not reported	ANCA antibodies were always negative and all responded to steroids.	25%
**Mamlouk O. et al. (2019)** [57]	Observational study	16	0.07%	Glomerulopathies were associated acute tubulointerstitial nephritis (ATIN) without glomerulonephritis and nine cases of ATIN with glomerulopathies. CPI were discontinued and steroids given. For AKI > grade 2 or proteinuria >1 gram/day, kidney biopsy should be performed.	93%
**Kitchlu A. et al. (2020)** [55]	Systematic review	45	Not reported	Most frequent manifestations were pauci-immune GN and renal vasculitis (27%), followed by podocytopathies (minimal change disease MCD; 20%) and C3 GN (11%).	88%
**Cortazar F.B. (2020)** [53]	Observational study	138	Not reported	AKI occurred at a median time of 14 weeks, grade 3 in about 57% and requiring renal replacement therapy in 9% with persistent renal damage in 15%. At rechallenge with CPI, recurrence rate was of 23%. Risk factors include use of PPI, lower eGFR at baseline and concomitant anti-PD1 and anti-CTLA4 therapy. Renal biopsy should be always performed.	3% at diagnosis
**Gupta S. et al. (2021)** [54]	Observational study	429	Not reported	AKI occurred mostly after 16 weeks from CPI. Lower baseline eGFR, PPI use and prior or concomitant extrarenal IRAEs were associated. In 60% of cases there were concomitant kidney toxic drugs. 5% of patients required other immunosuppressive therapy and 7% received renal replacement.	10%

AKI: acute kidney injury, PPI: proton-pump inhibitor, NSAID: Nonsteroidal anti-inflammatory drug, ANCA: Antineutrophil Cytoplasmic Antibodies, ATIN: Acute tubulointerstitial nephritis, GN: glomerulonephritis, MCD: minimal change disease, eGFR: Estimated Glomerular Filtration Rate.

### 3.6. Cardiovascular Toxicities

Some reports of cardiological CPI-related events have been published in the last few years (Table 7). As mentioned above, myocarditis is a life-threatening IRAE, has a median incidence of 0.27–1.14% [59,60] and develops after one month from the beginning of CPI treatment, particularly anti-PD1. Nearly half of all myocarditis leads to a Major Adverse Cardiac Event (MACE), defined as a composite of cardiovascular death, cardiac arrest, cardiogenic shock and hemodynamically significant complete heart block [61].

Regarding predictors of cardiac complications, Awadalla et al. [62] documented that a lower global longitudinal strain (GLS) of left ventricle by echocardiography was strongly associated with MACE. In particular, each reduction of 1% in GLS was associated with a 1.5-fold increase in MACE among cases with reduced ejection fraction and a 4.4-fold increase in those with a preserved ejection fraction. Additionally, Mahmood and colleagues [61] reported a threshold of 1.5 ng/dL troponin T levels as a predictor of MACE with 95% specificity.

Two other possible complications of CPI are to be mentioned: pericardial effusion/inflammation and vasculitis, particularly Horton arteritis. As reported by Salem and colleagues [60], the former was mainly observed in lung-cancer patients, whereas the latter was in those with melanoma. In conclusion, if cardiac involvement is suspected, attention to echocardiographic findings and the monitoring of heart-damage biomarkers should be performed to establish prompt intervention and reduce the risk of MACE. As in hematological and neurological IRAEs, CPI should be discontinued, and immunosuppressive therapy should be started.

**Table 7 pharmaceuticals-15-00557-t007:** Cardiovascular immune-related AE after checkpoint inhibitors (CPI).

References	Type of Study	Patients	Frequency	Main Findings
**Mahmood S.S. et al. (2018)** [61]	Observational study	35	1.14% for myocarditis 0.52% for MACE	Cardiovascular IRAEs were more common with combination therapy (anti-PD1 + anti-CTLA4 inhibitors), with median onset of 34 days. Higher level of troponin was detected at admission in nearly all patients. Treatment with high doses of steroids was associated with reduced incidence of major cardiologic events.
**Salem JE et al. (2018)** [60]	Observational study	31,321 evaluated records	Not reported	Higher incidence of myocarditis, pericardial diseases, supraventricular arrhythmias and vasculitis was described after CPI versus the general population. The median time to onset was of about 30 days. Epidosed were mainly severe (>80%), with a mortality of 50% for myocarditis.
**Hu J. et al. (2019)** [59]	Systematic Review	Not reported	0.27–1.14% of myositis No data for pericarditis	Most frequent cardiovascular IRAEs were myocarditis, pericardial diseases and vasculitis. Patients receiving CPI had 11-fold increase of myocarditis compared with the general populations.
**Awadalla M et al. (2020)** [62]	Observational study	101	Not reported	Global longitudinal strain (GLS) at echocardiography did not predict overall cardiac IRAEs but identified patients at a higher risk of MACE.

MACE: major adverse cardiac event, ECG: electrocardiography, GLS: global longitudinal strain.

## 4. Conclusions

The deepening knowledge of tumor biology and its microenvironment is maximizing the use of biological drugs in oncological and hematological diseases. These include CPIs that are aimed at restoring immune surveillance against tumors. In comparison with traditional chemotherapy, CPI-related adverse events are drastically different, mainly immune mediated and broadly involving several organs and tissues. Clinicians are continuously learning how to detect, treat and possibly foresee them. Unfortunately, laboratory/instrumental predictors are lacking, although active research is ongoing, and suspicion relies mainly on clinical observation and monitoring of blood counts, liver and kidney tests, hormone, neurologic and cardiac status.

In this setting, hematologic IRAEs represent a good example, given their broad differential diagnosis in cancer patients, possibly leading to a dramatic diagnostic and therapeutic delay resulting in fatal outcome. Along with AIHA, other hematological, neurological and cardiological IRAEs may also be life-threatening and require the prompt discontinuation of the CPI and institution of immunosuppressive therapy and supportive measures.

Additionally, these very severe forms, along with endocrine dysfunctions, are more likely to leave permanent organ damage and to become chronic, and thus predictors are even more important. At variance, cutaneous and gastroenteric IRAEs, although very common, are generally milder, and the discontinuation of CPI is usually required in severe cases only. Renal IRAEs are somewhat in the middle of the spectrum, since they are rare and potentially reversible, and early histologic findings may aid in the differential diagnosis to inform the utility of CPI discontinuation.

## Figures and Tables

**Figure 1 pharmaceuticals-15-00557-f001:**
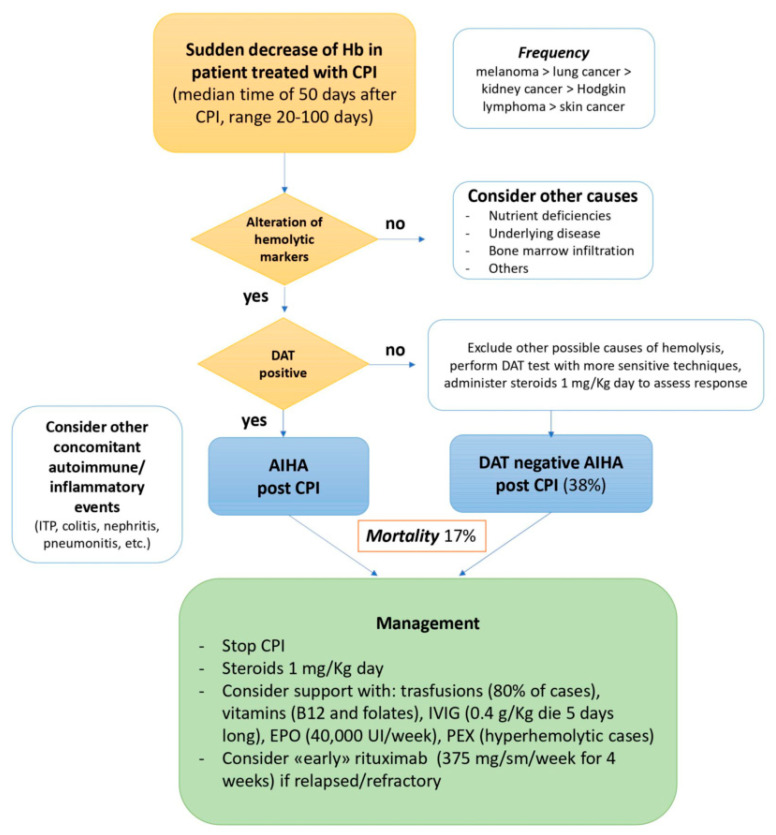
Diagnosis and management of patients affected by autoimmune hemolytic anemia (AIHA) after checkpoint inhibitors (CPI). DAT direct antiglobulin test; ITP immune thrombocytopenia; IVIG intravenous immunoglobulin; EPO recombinant human erythropoietin; PEX plasma exchange.

**Figure 2 pharmaceuticals-15-00557-f002:**
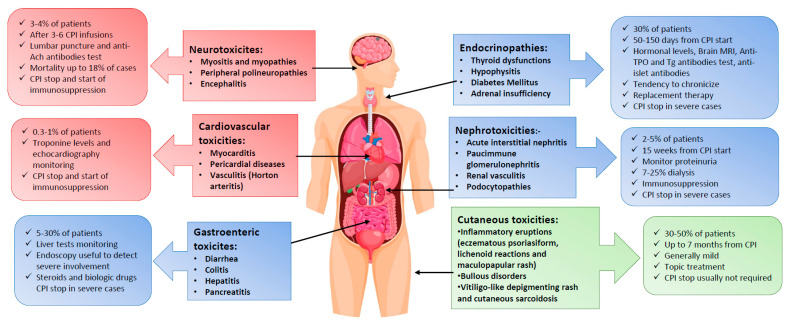
Extra-hematologic immune-related adverse events (IRAEs) after checkpoint inhibitors (CPI).

## Data Availability

Data is contained within the article.

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
