# Peer review of "Hematological and Extra-Hematological Autoimmune Complications after Checkpoint Inhibitors"

_pharmaceuticals, 2022, doi:10.3390/ph15050557_

Round 1
Reviewer 1 Report
Well written and well organised. Very comprehensive review.
Author Response
We thank the Referee for revising our manuscript and for the positive feedback.
Reviewer 2 Report
In this review the authors describe hematological and extra-ematological autoimmune complications after checkpoint inhibitors.
I suggest some changes:
- 1.1 Autoimmune hemolytic anemia instead of 2.2 (line 72)
- 1.2 Other hematological toxicities instead of 2.2 (line 114)
- 2.1 Immune-related endocrinopathies instead of 3.1 (line 134)
- 2.2 Cutaneous adverse events instead of 3.2 (line 158)
- 2.3 Gastroenteric side effects instead of 3.3 (line 176)
- 2.4 Neuromuscolar complications instead of 3.4 (line 201)
- 2.5 Nephrotoxicity instead of 3.5 (line 221)
- 2.6 Cardiovascular toxicities instead of 3.6 (line 240)
- Conclusions without number
- in table 2-3-4-5-6-7 please replace "/" with not applicable or with another locution..
Author Response
Many thanks for revising our manuscript and for the important suggestions.
I suggest some changes:
1.1 Autoimmune hemolytic anemia instead of 2.2 (line 72)
1.2 Other hematological toxicities instead of 2.2 (line 114)
2.1 Immune-related endocrinopathies instead of 3.1 (line 134)
2.2 Cutaneous adverse events instead of 3.2 (line 158)
2.3 Gastroenteric side effects instead of 3.3 (line 176)
2.4 Neuromuscolar complications instead of 3.4 (line 201)
2.5 Nephrotoxicity instead of 3.5 (line 221)
2.6 Cardiovascular toxicities instead of 3.6 (line 240)
Conclusions without number
in table 2-3-4-5-6-7 please replace "/" with not applicable or with another locution.
We corrected the manuscript as suggested.
Reviewer 3 Report
Authors reported that hematological and extra-hematological autoimmune complications after checkpoint inhibitors.
Although this manuscript is potentially interesting, several issues arise.
- Tables 1-7
The underlying diseases are required.
Main findings should be classified without several sentences.
- Aplastic anemia in including pure red cell anemia should be stated.
- ITP should be stated.
- Hemophagocyte syndrome should be stated.
- The mechanism of hematological complication should be stated. There are several mechanisms in hematological complication.
- Figure 6. There is not only AIHA in hematological complications after checkpoint inhibitors.
- Prophylaxis for complications after CPI may be useful.
- Hematological data may be useful for early predicting for hematological complication after CPI.
Author Response
We thank the Referee for the thorough revision and for the ameliorative comments.
Tables 1-7 - The underlying diseases are required. Main findings should be classified without several sentences.
As suggested we summarized the main findings avoiding long sentences. However, most studies contained a miscellaneous of cancers and listing all of them would make the tables more crowded and is beyond the scope of this review.
Aplastic anemia in including pure red cell anemia should be stated. ITP should be stated. Hemophagocyte syndrome should be stated.
We detailed the various acronyms as suggested. Thank you.
The mechanism of hematological complication should be stated. There are several mechanisms in hematological complication.
As a matter of fact, haematological complications, particularly cytopenias may reckon several causes in cancer patients, including the inhibitory effect on haematopoiesis of cancer-associated inflammation, direct bone marrow invasion, the toxic effect of chemo- and radiotherapy, and the induction of an autoimmune attack against hematopoietic precursors and peripheral blood cells. All of them represent differential diagnosis of immune related adverse events after CPIs. The latter are thought to be the result of a tolerance break induced by the reactivation of T-cells by inhibition of immunologic checkpoints. The text has been updated as follows, and a reference added for HLH:
“Hematological toxicities have been more frequently reported during anti-PD1 administration and mostly described in the form of unilinear or bilinear cytopenias (Table 1) [8,9]. The latter may reckon several causes in cancer patients, including the inhibitory effect of cancer-associated inflammation on haematopoiesis, bone marrow metastasis, the toxic effect of chemo- and radiotherapy, and the induction of an autoimmune attack against hematopoietic precursors and peripheral blood cells. All of them represent differential diagnosis of IRAES, that are thought to result from a tolerance break induced by the T-cells activation after CPIs.”
“Given their autoimmune/autoinflammatory nature, these manifestations are thought to be due to CPI-mediated immune dysregulation. A presumptive hyperinflammatory state as result of inhibition of cytotoxic T-cell activity seems to be the background of CPI-related HLH [Kalmuk et al, 2020]. Additionally, concomitant medications such as metamizole [18] may induce cytopenias through idiosyncratic reactions as consequence of immune imbalance.”
Kalmuk J, Puchalla J, Feng G, Giri A, Kaczmar J. Pembrolizumab-induced Hemophagocytic Lymphohistiocytosis: an immunotherapeutic challenge.” Cancers of the head & neck; 2020;5:3. https://doi.org/10.1186/s41199-020-0050-3
Figure 6. There is not only AIHA in hematological complications after checkpoint inhibitors.
We agree, but we would prefer to keep only AIHA in the figure since it is the most frequent and has a more comprehensive diagnosis.
Prophylaxis for complications after CPI may be useful. Hematological data may be useful for early predicting for hematological complication after CPI.
We agree with the Referee and added some sentences to the section:
“Overall, such forms may be difficult to diagnose and often require the exclusion of other secondary forms, possibly delaying proper treatment. Differently from AIHA, where DAT is available for the diagnosis, other hematologic IRAES are diagnosed after observing a drop of blood counts or alteration of HLH markers (serum ferritin, triglycerides, cytopenias, organomegalies, fever, etc.).” (…) “Altogether, even if rare, these complications are potentially fatal and require high awareness and adequate clinical and laboratory monitoring to establish prompt therapeutic measures [17]. “